# Ability of Genomic Prediction to Bi-Parent-Derived Breeding Population Using Public Data for Soybean Oil and Protein Content

**DOI:** 10.3390/plants13091260

**Published:** 2024-04-30

**Authors:** Chenhui Li, Qing Yang, Bingqiang Liu, Xiaolei Shi, Zhi Liu, Chunyan Yang, Tao Wang, Fuming Xiao, Mengchen Zhang, Ainong Shi, Long Yan

**Affiliations:** 1College of Life Sciences, Hebei Agricultural University, Baoding 071001, China; 18317312502@163.com; 2Hebei Laboratory of Crop Genetics and Breeding, National Soybean Improvement Center Shijiazhuang Sub-Center, Huang-Huai-Hai Key Laboratory of Biology and Genetic Improvement of Soybean, Ministry of Agriculture and Rural Affairs, Institute of Cereal and Oil Crops, Hebei Academy of Agricultural and Forestry Sciences, High-Tech Industrial Development Zone, 162 Hengshan St., Shijiazhuang 050035, China; qyang0807008@163.com (Q.Y.); liubingqiang@aliyun.com (B.L.); shixiaolei59@163.com (X.S.); zhiliulin@sina.com (Z.L.); chyyang66@163.com (C.Y.); 3Handan Academy of Agricultural Science, Handan 056001, China; wt414210391@163.com (T.W.); 13930083220@163.com (F.X.); 4Department of Horticulture, University of Arkansas, Fayetteville, AR 72701, USA

**Keywords:** soybean, protein content, oil content, GP, prediction ability, G-BLUP

## Abstract

Genomic selection (GS) is a marker-based selection method used to improve the genetic gain of quantitative traits in plant breeding. A large number of breeding datasets are available in the soybean database, and the application of these public datasets in GS will improve breeding efficiency and reduce time and cost. However, the most important problem to be solved is how to improve the ability of across-population prediction. The objectives of this study were to perform genomic prediction (GP) and estimate the prediction ability (PA) for seed oil and protein contents in soybean using available public datasets to predict breeding populations in current, ongoing breeding programs. In this study, six public datasets of USDA GRIN soybean germplasm accessions with available phenotypic data of seed oil and protein contents from different experimental populations and their genotypic data of single-nucleotide polymorphisms (SNPs) were used to perform GP and to predict a bi-parent-derived breeding population in our experiment. The average PA was 0.55 and 0.50 for seed oil and protein contents within the bi-parents population according to the within-population prediction; and 0.45 for oil and 0.39 for protein content when the six USDA populations were combined and employed as training sets to predict the bi-parent-derived population. The results showed that four USDA-cultivated populations can be used as a training set individually or combined to predict oil and protein contents in GS when using 800 or more USDA germplasm accessions as a training set. The smaller the genetic distance between training population and testing population, the higher the PA. The PA increased as the population size increased. In across-population prediction, no significant difference was observed in PA for oil and protein content among different models. The PA increased as the SNP number increased until a marker set consisted of 10,000 SNPs. This study provides reasonable suggestions and methods for breeders to utilize public datasets for GS. It will aid breeders in developing GS-assisted breeding strategies to develop elite soybean cultivars with high oil and protein contents.

## 1. Introduction

Soybean (*Glycine max* (L.) Merr.) is an economically important crop globally, with its yields contributing to nearly 60% of the world’s oilseed production as of 2023 (SoyStats2024, http://soystats.com/international-world-oilseed-production/) (accessed on 6 January 2024). The soybean serves as a vital source for various purposes, including human consumption, poultry and livestock feed, industrial application, and more. The soybean seeds are particularly notable for their high oil and protein content, comprising approximately 20% oil and 40% protein. This nutritional composition positions soybean as a crucial source of vegetable-derived protein, accounting for 60% of the globally vegetable-derived protein consumption, and vegetable oil, contributing to 29% of overall vegetable oil consumption [1]. 

Soybean seeds are renowned for their high protein and oil content, which are quantitative trait-controlled by multiple genes, influenced by both genetic and environmental factors [2,3]. Soybean breeding, traditionally, involves the creation of new genetic variation through controlled hybridization of two or more parents by phenotype selection to select offspring with desirable agronomic traits. The phenotypic selection is usually affected by environmental conditions and has large variation and changes according to different locations and years. Therefore, phenotyping needs to be performed by multiple locations and years. It makes phenotyping time-consuming and costly. However, molecular-assisted selection (MAS) offers a promising approach for soybean breeders to accelerate the development of new and improved plant cultivars with desirable traits. By leveraging molecular markers associated with target traits, MAS enables the more efficient selection of plants with desired genetic profiles, thereby accelerating the breeding process and reducing the reliance on labor-intensive and environmentally influenced phenotypic evaluations.

With advancing high-throughput genotyping methods and the rapid advance of sequencing technologies, a vast array of molecular markers is now available to aid soybean breeding. Quantitative trait locus (QTL) analyses have been used for mapping various traits using different types of molecular markers [4,5,6]. Among these markers, single-nucleotide polymorphism (SNP) has emerged as a cost-effective option, providing a large number of markers for conducting genome-wide association studies (GWASs) in plants [7,8,9]. In soybean breeding, significant efforts have been directed towards understanding the genetic basis of oil and protein content through QTL analyses for decades. For instance, QTL mapping and GWAS have identified SNP markers associated with oil and protein contents. Li et al. identified 31 SNPs located on 12 of the 20 chromosomes in soybean which were significantly associated with seed oil and protein contents [10]. These findings underscore the potential of SNP markers in facilitating the genetic improvement of soybean oil and protein contents through MAS and genomic selection (GS) approaches.

While MAS has been widely used in crop breeding [11,12,13], its applicability in crop breeding programs is often limited, especially for complex traits controlled by multiple genes with minor effects [14,15]. However, genomic selection (GS) through genomic prediction (GP) offers a promising solution for selecting such complex traits. Meuwissen et al. [16] proposed a GS method based on a genome-wide strategy, which utilizes genome-wide molecular markers and phenotype data (training population, TP) to effectively estimate the effects of all loci. This approach allows for the computation of the genomic estimated breeding value (GEBV), enabling the accurate prediction for the breeding population (BP) using only marker data. The selection of individuals in the BP can then be based solely on their GEBVs, without requiring phenotypic data. This reliance on genotypic information accelerates breeding cycles, reduces time and cost, and enhances the rate of genetic gain [17,18]. Alexandra et al. [19] evaluated the accuracy of GS for seed protein content and yield in a soybean breeding population, highlighting its potential in the soybean breeding program. Similarity, Zhang et al. [20] accessed the prediction ability (PA) of GS and MAS for seed weight in a population of 309 soybean germplasms, using 31,045 SNPs. A fundamental requirement for GP is that at least one molecular marker is in linkage disequilibrium (LD) with the QTL associated with the target trait. GP models leverage genetic relationships among individuals and information from LD between markers and QTL to make accurate predictions [21].

Indeed, GP holds great promise as an efficient method for improving quantitative traits, yet its accuracy is affected by various factors, such as marker density; population size; genetic architecture of the trait; LD between markers and QTLs; and relatedness between the TP and BP, model selection, and more [22,23,24]. With larger TP sizes, the accuracy of the molecular marker effect estimation by the model improves, leading to an enhanced PA. In general, increasing the number of molecular markers can effectively improve the PA as well [25,26,27]. A range of statistical models have been utilized in GP, including ridge regression best linear unbiased prediction (RR-BLUP) [28,29], genomic best linear unbiased prediction (G-BLUP) [30,31], Bayesian models [32,33], and machine learning models [34,35]. These statistical models play a crucial role in determining the PA in GP breeding programs [36,37,38]. Furthermore, the composition of the TP in relation to the BP is a critical factor for achieving high PA in GS breeding programs. The degree of genetic relationship between the TP and BP has been identified as a key factor influencing the effectiveness of GP [39,40,41,42]. Thus, careful consideration of TP composition and genetic relatedness is essential for maximizing the PA in breeding programs.

The implementation of GP breeding program demands extensive data, which are time-consuming and labor-intensive to collect. However, many plant breeding programs have accumulated large historical datasets of phenotypic information over the years. These datasets provide a valuable resource that can be integrated into GP methods, thereby enhancing genetic gain and reducing breeding cycle time [43,44]. There is an increasing interest in leveraging these rich historical breeding data with genotypic information so that breeders can enhance the accuracy and efficiency of GP methods. For instance, historical breeding test data have been effectively utilized in breeding programs such as the International Wheat Breeding Program [45] and the Rye (*Secale cereale* L.) Breeding Program [46] to improve GP performance. Integrating historical phenotypic data into GP approaches enables breeders to capitalize on valuable information accumulated over years of breeding efforts. This integration facilitates the development of more robust and efficient breeding strategies, ultimately accelerating the process of developing improved crop varieties.

The application of genomic prediction (GP) has demonstrated success in predicting essential agronomic traits and seed quality in soybean, encompassing traits like yield, plant maturity, protein, and oil content [19,20]. However, there remains a paucity of studies in soybean that harness historical breeding data for GP. Several questions persist regarding the viability of across-population prediction in soybean breeding endeavors, including whether the PA of across population prediction can be enhanced through the strategic selection of TP, choice of modeling approaches, and SNP marker densities. Hence, the primary objectives of this study were to leverage public datasets of USDA GRIN soybean germplasm accessions, featuring available phenotypic data on seed oil and protein contents, alongside genotypic SNP marker data. Our goal was to conduct GPs across various experimental populations, aiming to predict a bi-parent-derived breeding population in our experiment. We aimed to assess how different factors, such as the composition of the training set with varied genetic backgrounds and population structure, genetic relatedness to the testing set, different GP models, and SNP numbers, influence the GP performance for soybean oil and protein contents. By systematically investigating these factors, our study sought to provide insights into optimizing GP strategies for soybean breeding programs, ultimately enhancing the efficiency and effectiveness of genomic selection in improving soybean traits.

## 2. Results

### 2.1. Phenotypic Analysis

The phenotypic parameters of oil and protein contents in the six USDA populations and the breeding population derived from JD12 and NF58 exhibited a wide range (Table 1), indicating substantial variability conducive to genetic analysis. ANOVA revealed showed high broad-sense heritability for oil content (r = 0.94) and protein content (r = 0.93) in the breeding population (Table 2). Significant differences (*p* < 0.001) were observed in both oil and protein content among different environments, along with an extremely significant genotype–environment interaction variance (*p* < 0.001) in the breeding population. Furthermore, genotype variance within the breeding population reached highly significant levels (*p* < 0.001), indicating significant notable differences among genotypes within the population. Through a genetic correlation analysis of six USDA populations and the breeding population, the results indicate that the phenotype correlation between oil and protein content reached significant levels in seven populations, with values of −0.71 (A population), −0.75 (B population), −0.65 (C population), −0.50 (D population), −0.38 (E population), −0.26 (F population), and −0.87 (breeding population).

### 2.2. PCA and Phylogenetic Analysis

As described in the Materials and Methods section, the 4141 accessions were categorized into six groups (A, B, C, D, E, and F) based on the experimental locations at which they were phenotyped and their soybean *Glycine* species. To understand the influence of genetic similarity and population structure between the training set and testing set on the PA, we employed PCA and phylogenetic analysis among the six USDA populations and the bi-parental breeding population. Both PCA and phylogenetic analysis using 39,681 SNPs divided the 4141 USDA soybean germplasm accessions into two clusters, labeled I and II (sub-populations) (Figure 1a,b). The four cultivated soybean populations (“Max” depicted in red in Figure 1a, and A, B, C, and D in Figure 1b) merged together as Cluster I. The two wild soybean populations, labeled “Soja” in blue in Figure 1a and denoted as “E” (E population) and “F” (F population) in Figure 1b, merged together as Cluster II. These results indicated the presence of two distinct sub-populations within 4141 USDA germplasm accessions, with cultivated types (A, B, C, and D) exhibiting a different genetic background from wild soybeans (E and F). The bi-parent-derived population consisted of 175 lines and two parents (labeled “BP” in green in Figure 1a and “breeding population” in Figure 1b), was grouped into Cluster I alongside cultivated soybean accessions, particularly close to B and A, indicating that the breeding population shared a close genetic base with B and A, followed by C and D, but is notably distant from wild soybean (E and F).

### 2.3. Genomic Prediction through Within-Population Prediction and Across-Population Prediction for Each Population in USDA Germplasm

The PA of soybean seed oil and protein contents were estimated through within-population prediction within each of the six USDA populations. The PA was 0.86, 0.79, 0.70, 0.70, 0.70, and 0.51 for population A, B, C, D, E, and F, respectively, with averaged 0.71 for oil content; and 0.73, 0.66, 0.67, 0.67, 0.59, and 0.33 for population A, B, C, D, E, and F, respectively, and average 0.61 for protein content using G-BLUP model (Figure A1 and Table 3). These results demonstrate that the PA was high for both oil and protein contents, indicating that the efficiency of selecting the two traits in soybean breeding through GS by within-population prediction. 

The PA of soybean seed oil and protein contents was estimated through across-population prediction among the six USDA populations (Table 3). As demonstrated earlier, the six populations can be grouped into two clusters (I and II). Hence, we can categorize four types of PA by across-population prediction: (1) among the four cultivated populations (A, B, C, and D) within Cluster I; (2) among the two wild soybean populations (E and F) within Cluster II; (3) from Cluster I to Cluster II; and (4) from Cluster II to Cluster I. 

The PA (r−value) ranging from 0.21 to 0.77 averaged 0.55 for oil content; and from 0.16 to 0.52, it averaged 0.36 for protein content when performed across-population prediction among the four cultivated populations (A, B, C, and D) (Table 3). These results illustrate a wide range of variations in PA for both oil and protein contents, suggesting that the GS can effectively be utilized in soybean breeding programs to select for oil and protein contents through across-population prediction within cultivated soybean accessions. The PA was 0.16 and 0.37 from E to F, and 0.05 and 0.15 from F to E for oil and protein contents, respectively, by across-population prediction among the wild soybean populations (E and F) (Table 3). However, the PA exhibited very low r-values, close to zero, with a negative value in most cases, when performing across-population prediction from one cultivated population (either A, B, C, or D) to the wild species (either E or F), or vice versa (Table 3). These results indicate that the GS will not be efficient when conducting across-population prediction between cultivated soybean panels and wild soybean panels.

### 2.4. Genomic Prediction through Across-Population Prediction from Populations of USDA Germplasm Accessions to the Breeding Population

To estimate the effect of genetic similarity between the training set and testing set on the PA, the genetic distance among each of the six training sets (A, B, C, D, E, and F) and the breeding population was calculated (Table A2). When using each of the six USDA populations as a training set to perform GP for the bi-parental breeding population using the G-BLUP model, we observed variable PAs for either oil content or protein content. Smaller genetic distances were associated with a higher PA (r-value), whereas larger genetic distance corresponded to a lower PA (Table A2 and Figure 2a), suggesting that PA was related to the genetic distance between the training set and the testing set (breeding population) for both oil and protein contents in soybean. A linear model was constructed between PA (r-value) and genetic distance oil and protein content, respectively (Figure 2b), indicating that the genetic distance may be utilized to predict the selection efficiency. 

The PA was estimated for oil and protein contents based on the six USDA germplasm populations individually or combined as a training set to predict the bi-parent breeding population in this study. The average PA was 0.34 for oil content and 0.22 for protein content, respectively, while it ranged from −0.13 to 0.47 for oil content and from −0.18 to 0.45 for protein content in this study (Table 4). These results indicate that the different training sets can affect the PA, suggesting that the selection of the right training set is crucial during GP for genome breeding.

PA exhibited slight differences when the six public populations were integrated to form different training sets (Figure 3a). When only the closely related population A (genetic distance = 228.66; population size = 1007) was used, the PA was 0.37 for oil content and 0.39 for protein content. After integrating the second closely related population, B (genetic distance = 222.45, population size = 607), the PA was 0.36 for oil content and 0.33 for protein content. The highest PA of 0.47 for oil and 0.45 for protein content was achieved when the training set contained four cultivated populations; although it was not statistically significant, it rose by 0.02 and 0.06, respectively (Table 4). This suggests that either population A, B, C, and D individually or combined can be used to predict oil and protein contents in the breeding population. However, when the wild soybean population E or F was used as a training set to predict the breeding population in across-population prediction, the PA exhibited a negative r-value (Table 4), indicating that population E or F or their combination, EF, should not be used as a training set to predict oil and protein contents for other breeding populations.

PA was estimated for soybean seed oil and protein content by selecting different sizes of training sets from 400 to 4000 USDA accessions to predict the bi-parental breeding population (Figure 3b). When 400 accessions were used as a training set, the PA for oil and protein was 0.33 and 0.28, respectively. However, when 800 or more (800 to 4000) accessions were used as the training set, the PA showed a similar value for oil (~0.47) and protein (~0.44) (Figure 3b). This represents an increase of 0.14 and 0.16, respectively, compared to using 400 accessions as the training set, indicating that 800 or more accessions may be used as the training set to predict oil and protein contents in GS.

Regarding the training set, the four USDA cultivated populations (A, B, C, and D) can be used as a training set individually or combined to predict oil and protein contents in GS when using 800 or more USDA germplasm accessions as the training set.

### 2.5. Genomic Prediction between Across-Population with Different Model and Different SNP Number

The PA showed similar r-values when using G-BLUP, BRR, BL BB, and RR-BLUP models for either oil or protein content in across-population prediction. During across-population predictions, the training set contained the combined four cultivated populations (A, B, C, and D) to predict a bi-parent-derived breeding population (Figure 4a). The results indicate that the choice of model has little effect on PA for oil and protein contents in GS.

PA was estimated for soybean seed oil and protein contents through the four cultivated populations (A, B, C, and D) to predict a bi-parent-derived population with nine SNP sets from 100 SNPs to all 39,681 SNPs randomly selected using G-BLUP in this study (Figure 4b). The results showed that the different PA values (r-values) were observed, and the PA increased when a training set had more SNP numbers. When a set of 100 randomly selected SNPs was used, the PA was 0.13 and 0.08 for oil and protein content, respectively. However, when a SNP set consisted of 10,000 SNPs or more (from 10,000 to 39,861 SNPs) was used to perform prediction, the PA showed a similar value for oil (~0.46) and protein (~0.43) (Figure 4b). Additionally, we also observed that the PA (r-value) showed a similar value for oil and protein content, respectively, when using 10,000 or more (20,000 and 30,000) SNPs for cross (within-population) prediction among the four cultivated populations of USDA germplasm using the G-BLUP model (Table A3), indicating that 10,000 or more SNPs may be included to predict oil and protein contents in GS by either within- or across-population prediction.

### 2.6. Comparing Genomic Prediction Ability between within Breeding Population and Across-Population Prediction from Populations of USDA Accessions to the Breeding Population

The average PA (r-value) was 0.55 for oil content and 0.50 for protein content estimated by 10-fold cross-prediction within the bi-parent-derived population (Figure 5a). The observed breeding value was positively correlated with the estimated breeding value (Figure 5b,c). Within-population prediction exhibited higher PA than across-population prediction for oil and protein content when used G-BLUP model in this study (Table 3, Figure 5b,c). When the six populations from USDA germplasm were employed as a training set to predict the bi-parent-derived population, the PA was 0.45 for oil and 0.39 for protein content (Figure 5b,c). Although both PA values, either for oil or for protein content, showed a decreased of around 0.1 by across-population prediction, the PA was equal to or greater than 0.39, indicating that we can use public datasets to predict breeding populations such as the bi-parent-derived breeding population in this study in soybean breeding programs to select breeding lines and plants with high oil and protein contents through GS.

## 3. Discussion

With the advancement of sequencing technology, acquiring genotype data has become increasingly affordable. A vast amount of soybean sequencing data and phenotyping data in databases are available for GS, which can be applied in plant breeding practices to create greater economic value. However, there still exists a gap between within-population prediction ability and across-population prediction ability. Therefore, the key question is how to utilize these existing public data to predict breeding populations and improve ability of GS.

### 3.1. Genetic Similarity and Population Structure between Training Set and Testing Set and Training Set Size Has a Large Influence on the PA

PCA and population structure between the training set and testing set had effects on the PA estimation for oil and protein contents in this study. The public populations A, B, C, D, E, and F were divided into two sub-populations (Group I) (Figure 1), where A, B, C, and D merged together as one cluster, and E and F were merged into another (Group II). Our bi-parent-derived breeding population was merged to Group I based on population structure and phylogenetic analysis. Our results showed that the smaller the distance between the training set and testing set, the higher the PA in either within- or across-population prediction (Figure 2a; Table A2). The linear model between PA (r-value) and genetic distance showed this trend for oil and protein content through across-population prediction among the six USDA populations (Figure 2b). Similar results were reported by Beche et al. [47], showing that the closer the relationship between families, the higher the PA, when compared to non-related bi-parental populations. Riedelsheimer et al. [40] conducted a study using five bi-parental doubled-haploid maize populations developed from crosses involving four parents. They reported that the average accuracies close zero or even negative values when predicting individuals in bi-parental families based on data from unrelated bi-parental families. However, one of the most important steps in applying genetic relationship data to genetic breeding may be to analyze both training set and testing set for their population structure and genetic backgrounds, and thus to improve PA and accelerate breeding progress. The effect of genetic relationship on PA is mainly due to differences in allele frequencies that affect the accuracy of marker effect estimates, which are difficult to assess with statistical models [42,48,49]. Therefore, considering the influence of genetic similarity between the training set and the testing set on the PA is important for improving the PA. Breeders should select a training set from the material with a similar genetic background to the testing set.

In this study, we observed that the PA increased, but it was not significant for oil (0.36–0.47) and protein (0.39–0.45), as multiple populations were added into the training set (Figure 3b). Ten training sets were used to estimate PA, from 400 to 4000 USDA germplasm accessions, selected from the 4141 accessions with closest genetic distance to the breeding population to perform GP. As the size of TP increased, the PA of oil and protein content demonstrated a tendency toward increasing first when Np = 400–800, and then remained constant (Np = 800–4000) (Figure 3b). Many reports have shown that the accuracy of genome-wide prediction increases with the increase in population size. Previous GS studies have focused on the effect of the training set size on PA in within-population prediction and rarely considered the effect of training set size on PA in across-population prediction. Past and present results can provide some guidance for conducting GS-assisted breeding programs. Zhang et al. [50] investigated the effects of TP size on the PA of GS when a consistent number of 62 accessions were randomly assigned as a testing set, and the PA for GS decreased along with the reduction in the TP size. Similar results were reported that the size of the training set affected PA values [51]. Liu et al. [52] reported that the PA leveled off when the size of the TP was 14 times as the size of the BP. In contrast, the TP gradually leveled off when the size of the TP was three times the size of the BP. The reason for this difference may be due to target traits and the genetic relationship between the training set and testing set. 

As mentioned above, both genetic relationship and population size affect PA. In the current study, the kinship between the six USDA populations and breeding population was calculated. The genetic distance was sorted from smallest to largest: B (607), A (1007), C (965), D (811), E (508), and F (243) (Table A2). Each of the six USDA populations was used as a training set to predict the PA for breeding population, respectively. The PA was ordered from highest to lowest: A, B, C, D, E, and F (Figure 2a). This result may be due to the fact that both the kinship and training set size affect the PA values. The highest PA was observed when the populations A, B, C, and D combined as a training set to predict oil and protein contents in the breeding population. Compared to the six USAD populations (A, B, C, D, E, and F populations) combined as a training set (Figure 3a; Table 4), the PA of oil and protein contents increased by 0.02 and 0.06, respectively, than by one population as TP, but the increase was not significant. However, it decreased slightly compared to using four populations (A, B, C, and D) combined. As the distantly related population joined, the PA decreased. The results of the prediction showed that, in addition to considering the effect of the kinship on the PA, we also had to consider the effect of the size of the training set on the PA. In this study, public data were utilized as a training set to improve the ability of predicting bi-parental populations by considering the size of the training set and kinship between the training set and the testing set. This further demonstrates that it is possible to apply public data to GS breeding. Breeders should consider collecting and large amounts of breeding data and applying them to future GS studies and GS-assisted breeding.

### 3.2. Effect of Model Selection on Prediction Ability

In across-population prediction, all tested models had similar PA values without significant difference (Figure 4a). Smallwood et al. [53] demonstrated that Bayes B and/or G-BLUP were preferred for soybean fatty acid, protein, and oil contents in soybean. Riedelsheimer et al. [40] did not find significant differences between RR-BLUP, BL, and other models in predicting multiple traits. In previous studies, such as that by Kaler et al. [39], different PA values had been observed when different models were used. Moreover, BL model showed better results than BLUP model [54]. The reason for the different results may be that the PA is highly dependent on the precise estimation of marker effects by the statistical models. The estimation of marker effects is influenced by the allele frequency of each locus across the entire genome and varied among populations [55]. Moreover, Alexandra [19] points out that the PA was further improved by including epistasis in the GBLUP model in soybean seed protein content. Plant breeding programs often have access to a large amount of historical data that are highly unbalanced, particularly across years. Therefore, for multi-year and multi-location trials, developing suitable models that account for genotype-by-environment interactions can effectively enhance the PA of GS [56,57].

### 3.3. Effect of Markers Number Selection on Prediction Ability

When the training set contained the combined four cultivated populations (A, B, C, and D) to predict a bi-parent-derived population with different SNP number subsets, the PA increased with the increasing marker number until an SNP set consisted of 10,000 SNPs reached a maximum in PA (Figure 4b and Table A3). When using a different number of SNPs (100, 1000, 10,000, 20,000, and 30,000) for across-population prediction among the four cultivated populations of USDA germplasm, 10,000 or more SNPs showed a similar PA value for oil and protein (Table A3). Zhang et al. [20] reported a PA of 0.85 when a set consisted of 2000 or more SNPs was used, remaining 0.80 until the set included 500 or more SNPs for predicting seed weight in soybean. Liu et al. [52] showed that the PA began to level off when the number of markers increased to 1000 and 7000 in bi-parental population and natural populations, respectively. In summary, when the PA reached a plateau, additional markers became largely redundant and did not further improve predictive ability. However, the number of markers required for the PA to reach a plateau varies from population to population [58,59]. This phenomenon can be attributed to the complexity of genetic structure and the varying levels of diversity among populations. The reason may be that nearby markers with significant genetic structure and high LD levels between them are necessary to ensure that at least one marker could capture a trait-associated locus within the LD [16,20]. Therefore, to ascertain the optimal number of markers for GS-assisted breeding, a program based on the above results could serve as a reference to reduce breeding costs. Additionally, besides considering the number of markers, many studies have incorporated significant loci into GS to enhance PA. Sarinelli et al. [60] demonstrated that adding markers associated with large-effect genes or QTL as fixed effects in the model increased the overall model PA for most training population sizes evaluated using historical unbalanced phenotypic data. Hence, Breeders should consider adding major genes as fixed effects to improve model PA. With the development of breeding and the increasing size of populations, marker effects can be estimated more precisely [61]. Breeders should collect more historical breeding data from multi-year, multi-environment trials and apply them to GS breeding to significantly enhance genetic gain and accelerate breeding outcomes.

### 3.4. The Within-Population Prediction Got Higher Ability Than Across-Population Prediction

High accuracies were achieved in within-population prediction for both oil and protein contents in this study. The average PA was 0.55 for oil content and 0.50 for protein content within the bi-parent-derived breeding population (Figure 5a). The average PA ranged from 0.51 to 0.86 (oil content) and from 0.33 to 0.73 (protein content) through within-population prediction within the six USDA populations (Table 4). Smallwood [53] reported a PA of 0.56 for oil content and 0.67 for protein content in a soybean population comprising 860 F5-derived RILs derived from the parental lines Essex and Williams 82. However, within-population prediction may have limited application in a breeding program, as it predicts individuals from same population with narrow genetic base. Across-population prediction, on the other hand, can predict GEBV for any population, making it more useful for plant breeders to use routinely generated data in a breeding program to make predictions for independent populations. Therefore, we evaluated the ability of GP through across-population predictions. When using the six USDA populations combined as the training set to predict the bi-parent-derived population, the PA was 0.45 (oil content) and 0.39 (protein content) (Figure 5a). The averaged PA was 0.17 (oil content) and 0.18 (protein content) when performing across-population prediction among the six USDA population (Table 3), indicating that the across-population prediction had lower accuracies than within-population prediction for both oil and protein contents. Alexandra et al. [19] compared GS ability between within- or across-population prediction protein content in soybean and reported that the within-population prediction was more efficient. Stewart-Brown et al. [62] reported that the average PA was 0.60 and 0.52 in the within-population prediction and 0.55 and 0.30 in across-population prediction for protein and oil content, respectively. Thavamanikumar et al. [63] also showed that PA based on ten-fold cross-validation in each population was generally higher than PA than using marker effects from one population to predict the traits in another population. Liu et al. [52] reported that moderate-to-high accuracies were obtained when predictions were made within populations; in contrast, across-population genomic prediction accuracies were very low. 

Considering environmental impacts when conducting GS is crucial for improving prediction accuracy. While the heritability of oil and protein content in the breeding population was high in this study, the phenotypic data were obtained from various years and locations within the public dataset, without accounting for environmental factors. Sarinelli [59] demonstrated that including an additional one-year evaluation for most lines in a common environment notably improved the PA for heading date by up to 8%. Therefore, it is essential to consider environmental impacts when conducting GS. Plant breeding programs often have highly unbalanced historical data, particularly across different years. Dawson et al. [45] aimed to explore methods of integrating GP with genotype-by-environment (G × E) interaction models to effectively target untested lines across different locations. Hence, breeders utilizing public data for GS should consider environmental influences to enhance prediction ability.

## 4. Materials and Methods

### 4.1. Plant Materials 

Two sources of soybean materials were used in this study: USDA GRIN soybean germplasm accessions and a bi-parent-derived breeding population. A total of 4141 soybean accessions have available seed oil and protein contents at USDA GRIN, and the data can be download from the following websites: https://npgsweb.ars-grin.gov/gringlobal/descriptordetail?id=51016 and https://npgsweb.ars-grin.gov/gringlobal/descriptordetail?id=51019 (accessed on 5 April 2023). The 4141 accessions were divided into six panels based on the experimental locations where they were phenotyped and the soybean *Glycine* species. The six soybean panels consist of four cultivated soybean panels (MAX IL 0102, MAX MS 9901, MAX MS 9697, and MAX IL 9495) and two wild soybean panels (SOJA IL 9899 and SOJA MS 9899), defined as the A, B, C, D, E, and F populations, respectively (Table A1).

In our laboratory, we utilized a bi-parent-derived population comprising 175 F_2:6_ lines, with the two parent lines serving as the testing set (referred to as the breeding population, BP) for performing GP in this study. The two parental soybean cultivars, JD12 and NF58, exhibit contrasting quality traits. JD12 is characterized by its high protein content, with oil content around 17% and protein content around 46%. On the other hand, NF58 is known for its high oil content, with oil content approximately 24% and protein content approximately 36%. The cross between the two cultivars was initiated in 2003, and the bi-parent-derived breeding population was subsequently developed through single seed descent (SSD), resulting in the generation of 175 F_2:6_ lines.

### 4.2. Phenotyping

The oil and protein contents in the 4141 soybean accessions belonging to the six populations (A, B, C, D, E, and F) were obtained (downloaded) from the USDA GRIN database. For soybean accessions with yellow seed coats, oil and protein contents were analyzed using the near-infrared reflectance method. For pigmented or mottled soybean accessions, the quantification of oil and protein contents was conducted using the Kjeldahl method and Butt extraction method, respectively. 

The 175 F2:6 breeding lines, along with their two parents, underwent filed experiments from 2011 to 2012 at Shijiazhuang, Cangzhou, and Handan in China. The phenotypic data for the bi-parent-derived population were collected using a randomized complete block design (RCBD) with two replications. The plants were grown in rows that were 2 m long and spaced 0.5 m apart. Each parental and RIL line was grown in three replications. Harvesting was performed when 95% of the pods in each accession reached maturity. Standard agronomic practices were followed to grow the soybean plants. After harvesting, phenotypic data for oil and protein content were collected by randomly selecting three plants from the middle row for each genotype at the maturity stage. It was calculated by taking the average for three plants selected from each replicate. Approximately 20 grams (g) of seeds was randomly selected from each line for the evaluation of protein and oil contents using a MATRIX-I (BRUKER, Berlin, Germany) NIR spectrometer. Best Linear Unbiased Estimations (BLUEs) of phenotype values across different environments were obtained using mixed linear models from the R package lme4. The BLUEs for the combined environment were estimated using the following model:Yijk = μ + Ei + Rj (Ei) + Gk + εijk
where Yijk represents the oil or protein content per plant, µ represents overall mean effect, Ei is effect of the ith environment, and Rj is the effect of the jth replicate. GK is the effect of the kth genotype; and εijk represents the effect of the error associated with the ith environment, jth replicate, and kth genotype

Descriptive analysis and Analysis of Variance (ANOVA) of trait phenotypes were conducted using SPSS 22.0 (IBM Corporation, Armonk, NY, USA). Broad-sense heritability estimates were calculated using the following formula: h2 = σ2G/(σ2G + σ2Ge/e + σ2*ε*/re), where σ2G, σ2Ge, and σ2*ε* represent genetic variance, genotype by environment interaction variance, and residual error variance, respectively; and e and r represent the number of environments and replicates, respectively. Phenotypic correlation analysis was calculated using the “pearson” method of “cor” function of the R language.

### 4.3. Genotyping

The SNP data of the 4141 accessions in the six populations (A, B, C, D, E, and F) are available with the Illumina Infinium SoySNP50K Bead Chip from the Soybean Genetics and Improvement Laboratory, USDA-ARS, Beltsville, MD, USA, and can be downloaded at https://www.soybase.org/snps/ (accessed on 5 April 2023) [64].

For the 175 F_2:6_ breeding lines and parents, we performed SNP genotyping using the same SoySNP50K Bead Chip [64]. DNA was extracted from the leaf tissue of each breeding line, including two parents, and genotyped with the SoySNP50K Bead Chip described by Song et al. [64]. A total of 51,335 SNPs were obtained across the 175 breeding lines and two parents. 

After combining both SNP sets from the 4141 accessions and the 175 breeding lines, along with two parents, and filtering out with minor allele frequency (MAF) < 5% and missing data > 10%, 39,681 SNPs were retained for further analysis in this study. Following the filtering process, missing SNP data across soybean genotypes were imputed using Beagle 5.4 [65] before performing GP analysis. 

### 4.4. Genomic Prediction

The PA was calculated as the Pearson correlation coefficient (r-value) between the observed oil/protein content value or BLUEs (best linear unbiased estimate) and GEBVs. Various models, including G-BLUP, Bayes B (BB), Bayesian LASSO (BL), Bayesian ridge regression (BRR), and rrBLUP, were employed to predict GEBVs. A ten-fold cross-validation approach [48,66,67] was implemented for GP analysis in all within-population predictions. This procedure involves randomly dividing the dataset into ten subsets, where 90% of the samples were used as the training set, and the remaining 10% as the validation set (testing set). PA was defined as the Pearson’s correlation coefficient calculated for each fold or testing set and then averaged over 10 folds to obtain the final PA value [68]. All statistical models were executed in R language version 4.3.0 (R Core Team 2023, https://www.R-project.org/ (accessed on 5 April 2023). The G-BLUP, BB, BL, and BRR models were fitted using the Bayesian generalized linear regression (BGLR) package [69]. RR-BLUP was fitted using the “mixed.solve” functions in the “rr-BLUP” package. All Bayesian approaches were run as single chains of 10,000 iterations, discarding the first 2000 iterations as burn-in, using the BGLR package.

### 4.5. Training Population Selection

This study employed several strategies for across-population prediction, including utilizing different training sets (TSs), varying training set sizes, and assessing the genetic relationship between the training set and testing set (in this case, the bi-parent-derived breeding population). 

To calculate kinship among the six USDA populations and the bi-parent breeding population, several analyses were conducted: (1) Principal Component Analysis (PCA) was performed to capture the underlying structure of the genetic variation among the populations. This analysis helps to visualize the phylogenetic relationships between each sample. (2) Genetic distance analysis was conducted to quantify the similarity or dissimilarity (distance) between samples or sub-populations. This information is crucial for understanding the genetic relationships between populations. (3) Cluster analysis was employed to further explore the relationships between populations (training population and testing population). This analysis helps to identify groups of samples or populations with similar genetic profiles. 

For the genetic analysis, SNP data were processed using Plink, where each SNP allele was coded 0, 1, or 2, representing the major allele, heterozygous allele, and minor allele, respectively, based on the number of copies of the minor allele. Genetic distance was calculated using the “Euclidean” method of the “dist” function of the R language. Cluster analysis was performed based on genetic distance by the “average” method of “hclust” function of the R language. All of these analyses, including PCA, cluster analysis, and genetic distance calculations, were implemented in R version 4.3.0. These methods collectively provide insights into the genetic relationships between populations and help inform the across-population prediction process.

Each population (A, B, C, D, E, and F) within the 4141 USDA germplasm accessions, as well as the bi-parent-derived breeding population, consisted of the 175 F2:6 line, along the two parents, and underwent GP using G-BLUP via ten-fold cross-validation. To investigate the relationship between PA and genetic distance, GP was conducted for both within-population prediction and across-population prediction in the six populations of the USDA germplasm. Furthermore, the six populations were aggregated as a training set to execute across-population prediction for the bi-parent-derived breeding population (175 F2:6 lines and two parents). A correlation analysis was performed to examine the relationship between the genetic distance and PA among the six public populations and the bi-parent-derived breeding population using the G-BLUP model. This analysis aimed to elucidate how genetic distance influences prediction accuracy across different populations.

PA was also assessed based on the size of the training set. In this study, ten training sets were utilized, ranging from 400 to 4000 USDA germplasm accessions, increasing by increments of 400 accessions each time. These accessions were selected from the total pool of 4141 accessions based on their genetic distance to the breeding population. Genomic prediction (GP) using G-BLUP was then conducted for each training set to evaluate the impact of the training-set size on prediction accuracy.

### 4.6. Marker Selection

GP was conducted using nine SNP sets, including 100, 300, 500, 1000, 3000, 5000, 10,000, 20,000, and 30,000 SNPs randomly selected from a pool of 39,681 SNPs. In each GP analysis, the training set contained four cultivated populations (A, B, C, and D populations) combined to predict GEVB in the bi-parent-derived breeding population. Additionally, across-population prediction was performed among the six sub-populations of USDA germplasm accessions using five SNP sets (100, 1000, 10,000, 20,000, and 30,000 SNPs) randomly selected from the 39,681 SNPs. Each GP analysis was repeated 100 times to ensure the reliability of the results. Mean and standard errors (SEs) corresponding to each SNP set were computed to evaluate the predictive performance across different sets. This rigorous approach aimed to provide robust insights into the effectiveness of different SNP-set sizes for GP.

## 5. Conclusions

In this study, we harnessed USDA soybean germplasm population datasets to predict the bi-parent-derived breeding population (BP) for soybean oil and protein content. Our findings underscored the importance of the genetic distance in determining the prediction ability (PA), with closer genetic proximity between the training sets and the BP resulting in a higher PA. Additionally, enlarging the training population size positively influenced the PA, with optimal results observed with a threshold of ≥800 germplasms or using four cultivated soybean populations from USDA, either individually or combined. Furthermore, our study revealed a consistent enhancement in PA with the increasing marker density, peaking when the number of markers reached 10,000. Interestingly, no significant difference in PA was observed among different models tested, indicating robustness across methodologies. The average PA for oil and protein content within the BP were 0.55 and 0.50, respectively. When utilizing six USDA populations as the training set to predict the BP, the PA for oil and protein content were slightly lower at 0.45 and 0.39, respectively. Overall, the study demonstrates the feasibility of utilizing public datasets for genomic selection (GS) breeding. We provide valuable insights and methods for breeders to effectively utilize public datasets, thereby facilitating the development of GS-assisted breeding strategies. By accelerating the breeding process and enhancing genetic gains in breeding programs, these findings contribute to the advancement of soybean breeding efforts.

## Figures and Tables

**Figure 1 plants-13-01260-f001:**
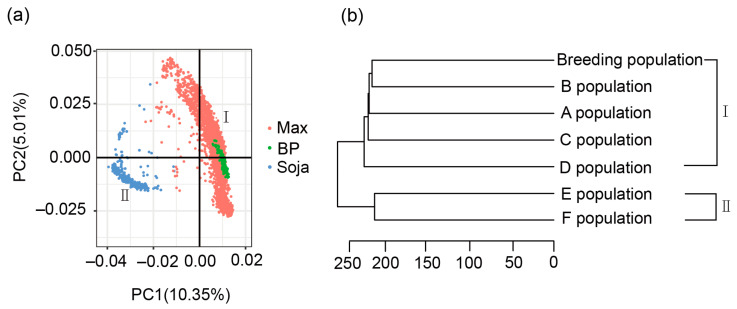
Principal Component Analysis (PCA) and phylogenetic tree of 4318 lines from 4141 USDA soybean germplasm accessions and 177 lines derived from a bi-parental breeding population. (**a**) PCA plot of the 4318 lines generated using R tools. Each dot represents an accession. Max (in red) indicates the cultivated soybean panels: populations A, B, C, and D. BP (green color) indicates the breeding bi-parent population. Soja (blue color) indicates wild soybean panels: population E and F. (**b**) Phylogenetic tree created using R tools for the seven populations: breeding population derived from a bi-parental population and the six sub-populations of USDA germplasm accessions, including four cultivated soybean populations (A, B, C, and D) and two wild soybean populations (E and F).

**Figure 2 plants-13-01260-f002:**
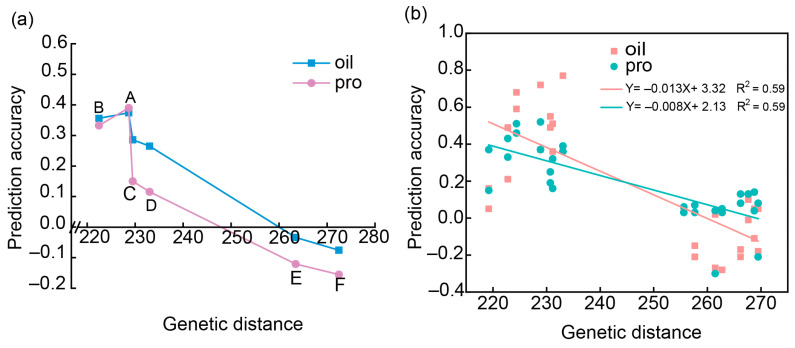
Relationship between genetic distance and prediction ability (PA) (r-value) of soybean seed oil and protein content was estimated. (**a**) Across-population prediction was employed through six USDA populations to predict the breeding population. Pink color indicates oil content, and blue content indicates protein content. A, B, C, D, E, and F represent each of the six USDA populations, respectively. (**b**) Linear regression between genetic distance and PA of oil (pink line) and protein (green line) content through across-population prediction using the populations in USDA germplasm accessions to predict oil and protein contents in the breeding population using G-BLUP model. Pink dots indicate oil content, and green dots indicate protein content.

**Figure 3 plants-13-01260-f003:**
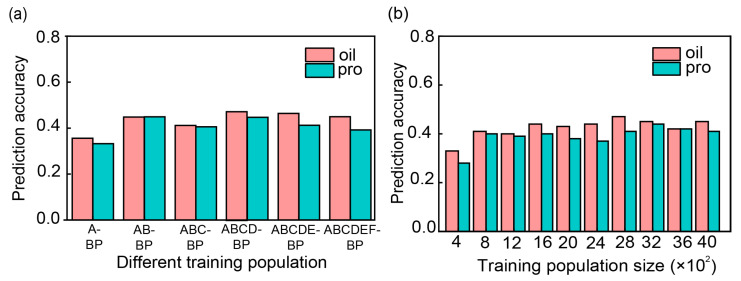
Prediction ability (PA) (r-value) was estimated for soybean seed oil and protein contents by selecting different training sets using G-BLUP model. (**a**) A-BP, AB-BP, ABC-BP, ABCD-BP, ABCDEF-BP, and ABCDEF-BP, respectively, represent different populations individually or combined from USDA population as training set to predict breeding population. (**b**) PA was estimated through selecting different sizes of training sets from 400 to 4000 USDA accessions to predict the bi-parental breeding population. Pink color indicates oil content, and green color indicates protein content.

**Figure 4 plants-13-01260-f004:**
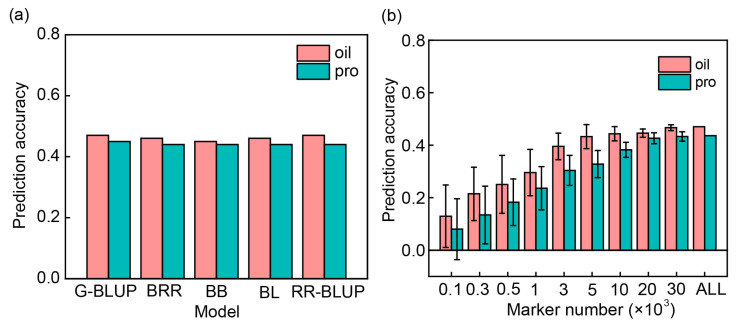
Prediction ability (PA) (r-value) of oil and protein content was estimated using the training set contained four cultivated populations (population A, B, C, and D) to predict bi-parent population by selecting different model and marker number. (**a**) PA was estimated using G-BLUP, RR-BLUP, BB, BL, and BRR models. (**b**) PA was estimated with nine SNP sets from 100 SNPs to all 39,681 SNPs randomly selected using G-BLUP. The SNP subsets were selected using random sampling. Prediction of randomly selected SNP sets was performed 100 times. Pink color indicates oil content, and green color indicates protein content.

**Figure 5 plants-13-01260-f005:**
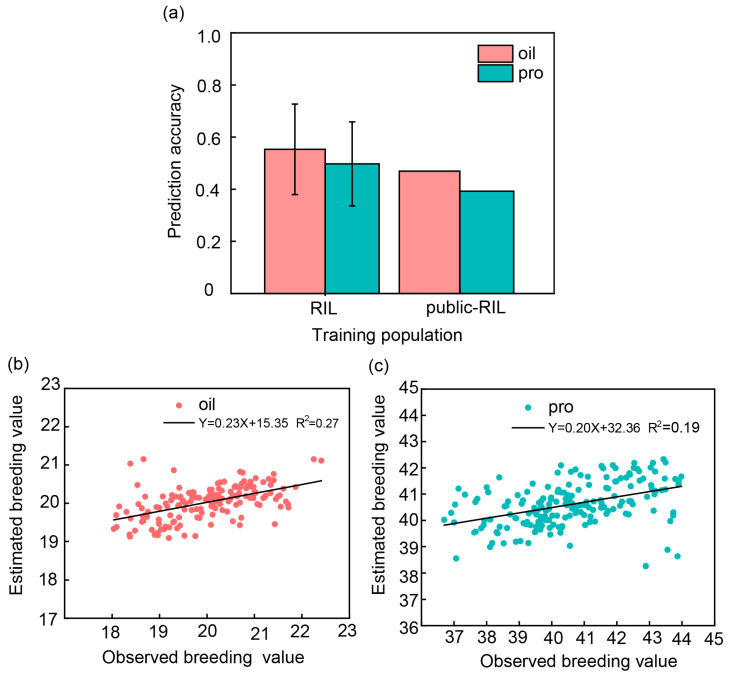
Prediction ability (PA) (r-value) was estimated for soybean seed oil and protein contents by within-population prediction and across-population prediction. (**a**) RIL indicates that the PA was estimated by 10-fold cross-prediction within the bi-parent-derived population, and public-RIL indicates that the PA was estimated by the six populations in USDA germplasm predicting the bi-parent-derived population using G-BLUP model. Error bars represent predicted value distribution of 10-fold cross-validation. Pink color indicates oil content, and green color indicates protein content. (**b**,**c**) PA was estimated by 10-fold cross-prediction within the bi-parent-derived population using G-BLUP. Pink color indicates the scatter plot of oil content (**b**); green color indicates the scatter plot of protein content (**c**); abscissa represents for observed breeding value; ordinate is for estimated breeding value; and the black line is for the trend line.

**Table 1 plants-13-01260-t001:** Descriptive statistical analysis of the protein and oil content of six USDA populations and the breeding population derived from JD12 and NF58.

Trait	Population	Min	Max	Average	SD	CV	Skewness	Kurtosis
Oil	MAX MS 0102: A	8.20	24.00	17.43	2.34	0.13	0.15	−0.09
	MAX MS 9901: B	8.20	21.70	17.42	2.07	0.12	−0.94	1.64
	MAX MS 9697: C	9.00	20.70	16.25	1.66	0.10	−0.41	0.51
	MAX IL 9495: D	11.2	25.40	18.45	2.19	0.12	−0.06	−0.05
	SOJA IL 9899: E	7.60	16.50	11.27	1.41	0.12	0.92	1.58
	SOJA MS 9899: F	7.50	14.70	10.86	1.04	0.10	−0.06	−0.1
	Breeding population	18.00	22.41	19.98	0.99	0.05	0.06	−0.76
Protein	MAX MS 0102: A	34.50	52.90	43.42	3.00	0.07	0.14	−0.21
	MAX MS 9901: B	37.40	56.30	45.52	2.75	0.06	0.39	0.75
	MAX MS 9697: C	35.10	57.40	46.08	2.31	0.05	0.20	1.54
	MAX IL 9495: D	37.70	55.40	45.84	2.65	0.06	−0.08	0.08
	SOJA IL 9899: E	35.50	55.30	45.71	3.37	0.07	0.24	0.41
	SOJA MS 9899: F	38.10	56.90	48.17	2.71	0.06	−0.13	0.08
	Breeding population	36.70	43.98	40.59	1.78	0.04	0	−0.71

**Table 2 plants-13-01260-t002:** Analysis of variance of oil and protein content and heritability estimates in the breeding populations derived from JD12 and NF58.

Population	Trait	Env. (E)	Genotype (G)	G×E	Error	*H_2_b*
Breeding population (BP)	oil	168.28 ***	37.05 ***	2.29 ***	0.38	0.94
	protein	203.44 ***	24.81 ***	2.11 ***	1.7	0.93

*p* < 0.001, indicated by ***.

**Table 3 plants-13-01260-t003:** Prediction ability (PA) (r-value) of soybean seed oil and protein contents was estimated through across-and cross-population prediction among the populations in six USDA populations and the bi-parental breeding population using G-BLUP model. Training sets are each of the six USDA populations: A, B, C, D, E, and F.

r-Value (Oil/Pro)	Testing Set						
Training Set	A	B	C	D	E	F	BP
							(Breeding Population)
A	0.86/0.73	0.59/0.46	0.43/0.37	0.64/0.36	−0.27/0.04	0.05/0.08	0.37/0.39
B	0.68/0.51	0.79/0.66	0.49/0.43	0.67/0.25	0.04/0.03	−0.01/0.13	0.36/0.33
C	0.72/0.52	0.21/0.33	0.70/0.67	0.51/0.32	−0.15/0.03	−0.21/0.13	0.29/0.15
D	0.77/0.39	0.49/0.19	0.36/0.16	0.69/0.63	0.03/0.03	0.05/0.14	0.27/0.12
E	0.02/−0.30	0.04/0.06	−0.21/0.07	−0.28/0.05	0.70/0.59	0.16/0.37	−0.08/−0.03
F	−0.18/−0.21	0.10/0.13	−0.17/0.08	−0.11/0.04	0.05/0.15	0.51/0.33	−0.12/−0.15

**Table 4 plants-13-01260-t004:** Prediction ability (PA) (r-value) was estimated for soybean seed oil and protein contents, where different populations individually or combined from the USDA population were used as training sets to predict the breeding population using the G-BLUP model.

USDA Germplasm Population	R-Value (Oil/Pro)
Training Set	BP as Testing Set (Breeding Population)
A	0.37/0.39
AB	0.45/0.45
ABC	0.41/0.41
ABCD	0.47/0.45
ABCDE	0.46/0.41
ABCDEF	0.46/0.39
B	0.36/0.33
BC	0.38/0.32
BCD	0.42/0.19
BCDE	0.42/0.26
BCDEF	0.41/0.26
C	0.29/0.15
CD	0.39/0.15
CDE	0.41/0.18
CDEF	0.38/0.17
D	0.27/0.12
DE	0.31/0.09
DEF	0.30/0.11
E	−0.08/−0.03
EF	−0.13/−0.18
F	−0.12/−0.15

## Data Availability

Publicly available datasets were analyzed in this study. These public data can be found here: https://npgsweb.ars-grin.gov/gringlobal/descriptordetail?id=51016 and https://npgsweb.ars-grin.gov/gringlobal/descriptordetail?id=51019 (accessed on 5 April 2023). Genotypic data of bi-parents-derived breeding population during the current study are available in the China National Center for Bioinformation (CNCB) repository, and the accession OMIX ID is OMIX005378. All the supporting data are available from the corresponding author upon reasonable request (dragonyan1979@163.com).

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
