# Peer review of "Ability of Genomic Prediction to Bi-Parent-Derived Breeding Population Using Public Data for Soybean Oil and Protein Content"

_plants, 2024, doi:10.3390/plants13091260_

Round 1

Reviewer 1 Report

Comments and Suggestions for Authors

Genomic selection has received a great deal of attention as a method to improve the efficiency of targeted crop breeding. In this paper, we examine the use of the USDA GRIN public dataset for genomic prediction of seed oil and protein content in soybean. This study is highly significant because the use of public databases has the potential to significantly reduce the amount of labor required for recent genetic research methods such as genomic selection and genome-wide association studies, which require large data sets. The study of various combinations of populations and models has been conducted, and this is commendable. However, there are some points that we consider insufficient, such as the explanation of the background and purpose of this study and the interpretation of the data obtained, and we therefore offer the following comments.

1.     Insufficient evaluation of the superiority or inferiority of the prediction ability of this study in comparison with previous studies

In this study, the combination of data sets, differences in the models employed, and the influence of the number of SNPs are examined using prediction ability as an indicator. These results are discussed and compared with previous findings (Discussion sections: 3.1, 3.2, and 3.3), which should help readers better understand the significance of this study. However, the reader may find it difficult to evaluate the results of this study because there is no discussion of the superiority of the prediction ability obtained in this study in comparison with previous studies (studies on components such as protein and oil, studies on soybean, and studies using actual measured data sets). A discussion of these perspectives would better inform readers about the usefulness and limitations of this study.

2.     Insufficient comparison with previous studies using public data

The most important idea of this study, using phenotypic data accumulated in breeding programs, is mentioned in the introduction (L105-113), but studies using publicly available historical data have been reported not only in the GP studies of wheat and Rye, but also in GWAS of rice and others have also been reported. I think that these past findings should be included to show readers the novelty of using GP in soybean.

Author Response

Dear Reviewer 2,

We greatly appreciate all your valuable comments and suggestions! We have thoroughly reviewed your feedback and have made revisions accordingly, which are highlighted in yellow throughout the manuscript. Furthermore, we have addressed the grammatical errors and improved the overall language quality of the manuscript. We believe that these revision have significantly enhanced the clarity and coherence of the paper. Thank you once again for your constructive feedback.

Please find below our responses to your comments, provided in a point-to-point manner!

  1. Insufficient evaluation of the superiority or inferiority of the prediction ability of this study in comparison with previous studies. In this study, the combination of data sets, differences in the models employed. and the influence of the number of SNPs are examined using prediction ability as an indicator. These results are discussed and compared with previous findings (Discussion sections: 3.1, 3.2, and 3.3), which should help readers better understand the significance of this study. However, the reader may find it difficult to evaluate the results of this study because there is no discussion of the superiority of the prediction ability obtained in this study in comparison with previous studies (studies on components such as protein and oil, studies on soybean, and studies using actual measured data sets). A discussion of these perspectives would better inform readers about the usefulness and limitations of this study.

Response: Thank you for your insightful comment. We acknowledge the importance of evaluating the predictive ability (PA) of our study in comparison with previous research. In our manuscript, we have provided a comparison of the PA of the GBLUP model for oil and protein content in soybean with values reported by Smallwood, demonstrating the utility of our findings (page 13: lines 479-481). However, we also recognize certain limitations in our study, particularly regarding marker and model selection, as well as the influence of environmental factors on quality traits. To address these limitations, we have incorporated discussions on these aspects into the manuscript (page 12: lines 438-443; page 13: 463-469, lines 503-514). We believe that by addressing these perspectives, readers will gain a better understanding of the significance and limitations of our study.

  1. Insufficient comparison with previous studies using public data

The most important idea of this study, using phenotypic data accumulated in breeding programs, is mentioned in the introduction (L105-113), but studies using publicly available historical data have been reported not only in the GP studies of wheat and Rye, but also in GWAS of rice and others have also been reported. l think that these past findings should be included to show readers the novelty of using GP in soybean.

Response: Thank you for your valuable suggestion. We acknowledge the importance of comparing our study with previous research utilizing publicly available historical data. Our study primarily focuses on leveraging existing public data to enhance genomic selection in breeding populations, as highlighted in the introduction (L129-135). This emphasis constitutes the novelty and superiority of our approach compared to previous studies (page 10: lines 360-362). Furthermore, it's worth noting that previous studies utilizing public data for genomic prediction have primarily focused on predictive ability (PA) within populations (page 3: lines 129-135). In contrast, our study extends this by utilizing public data to predict breeding populations, thus offering a novel contribution to the field. Once again, we appreciate your valuable comments and suggestions. We remain eager to hear any further feedback you may have.

Sincerely,

Chenhui Li

Reviewer 2 Report

Comments and Suggestions for Authors

Li et al used public datasets to perform genomic selection of oil and protein contents in soybean. Their aim was to implement GS-assisted breeding strategies to develop elite soybean cultivars. They used available data fro 453 soybean  accessions from USDA GRIN as the training set and 175 F2:6 lines, developed by the authors, and the two parental lines as the testing set. The article is interesting  and contribute with valuable insights into the deployment of GS strategies for enhancing oil and protein content in soybean.

Despite the quality of the english language that must be immproved, I have other comments:

1- How were the BLUEs calculated? What was the linear mixed model used?

2 - The genotype by environment interaction effect should be calculated. An analysis among environments should be performed, since you have used phenotypic data obtained from different years/locations

3 - The broad-sense heritability of the traits (oil and protein content) should be calculated.

4 - The components of variance relative to genotype variation and experimental error should be calculated.

5 - What is the genetic correlation for genotype responses across traits?

6 - Can you estimate the accuracy of the models?

7 - The validation/testing set is a population resulting from the cross of two cultivars with oil and protein contents not very contrasting (the values are more or less similar?). Also, the oil and protein content of this population was measured in one year only. These quality traits are described to be highly influenced by the environment... This should be discussed.

Comments on the Quality of English Language

The English language must be improved. Many sentences are not well written making it difficult to understand the text. A revision by an english native should be performed to improve the clarity and quality of the manuscript. I have highlighted with comments some examples in the text (pdf file attached) but all the text must be revised.

Author Response

Dear Reviewer,

We greatly appreciate all your valuable comments and suggestions! We have thoroughly reviewed your feedback and have made revisions accordingly, which are highlighted in yellow throughout the manuscript. Furthermore, we have rectified grammatical errors and improved the overall language quality of the manuscript, with revisions made by professors from American universities. We believe that these revision have significantly enhanced the clarity and coherence of the paper. Thank you once again for your constructive feedback.

Please find below our responses to your comments, provided in a point-to-point manner!

  1. How were the BLUEs calculated? What was the linear mixed model used?

Response: Thank you for your valuable suggestion! The BLUEs (Best Linear Unbiased Estimators) were calculated using mixed linear models implemented in the R package lme4. Specifically, we estimated the predicted means (BLUEs) for the combined environment using the following model:

Yijk = μ + Ei + Rj ( Ei ) + Gk + εijk

where Yijk represents the oil or protein content per plant, µ represents overall mean effect, Ei is effect of the ith environment and Rj is the effect of the jth replicate. GK is effect of the kth genotype; εijk represents effect of the error associated with the ith environment, jth replicate and kth genotype.

We have incorporated this description into the manuscript (see page 14, lines 554~562).

  1. The genotype by environment interaction effect should be calculated. An analysis among environments should be performed, since you have used phenotypic data obtained from different years/locations.

Response: Thank you for your insightful comments! We agree that analyzing “Genotype by environment interaction effects” is crucial, especially considering the diverse phenotypic data obtained from different years and locations. In our study, we conducted an Analysis of Variance (ANOVA) using SPSS 22.0 (IBM Corporation, America), which allowed us to examine genotype by environment interaction effects across multiple environments within the breeding population. The results revealed extremely significant genotype-environment interaction variance (P < 0.001) for both oil (2.29***) and protein (2.11***) content in breeding population. These findings are detailed on page 4, lines 153~155 and summarized in Table 2.

  1. The broad-sense heritability of the traits (oil and protein content) should be calculated.

Response: Thank you for highlighting the importance of calculating the broad-sense heritability for the traits under study. We have indeed calculated the broad-sense heritability of oil and protein content across two years and three locations within the breeding population. The heritability estimates were computed using the formula:

h22G / (σ2G + σ2Ge / e + σ2? / re)

where: σ2Gσ2Ge and σ2? represent genetic variance, genotype by environment interaction variance, and residual error variance, respectively; and e and r represent the number of environments and replicates, respectively.

The calculated broad-sense heritability were high for both oil (0.94) and protein (0.93) content in the breeding population. These results are detailed on page 4, lines 151-153, and further elaborated on page 14, lines 564-568, as well as summarized in Table 2.

We appreciate your suggestion, and we believe that including these heritability estimates strengthens the robustness of our findings.

4 The components of variance relative to genotype variation and experimental error should be calculated.

Response: We sincerely appreciate the valuable comments. We have calculated the components of genotype variation and experimental error across two years and three locations within the breeding population using SPSS 22.0 (IBM Corporation, America). Our analysis revealed extremely significant levels of genotype variance within the breeding population (P < 0.001). Additionally, the experimental error for oil and protein content was determined to be 0.38 and 1.70, respectively, across multiple environments within the breeding population. These findings are detailed on page 4, lines 156-158, and summarized in Table 2.

We believe that these analyses contribute to a comprehensive understanding of the variation observed in our study. Thank you once again for your valuable input.

  1. What is the genetic correlation for genotype responses across traits?

Response: Thank you for your inquiry. We calculated the genetic correlation for genotype responses across oil content and protein content using the "pearson" method of the "cor" function in the R language for both the breeding population and six USDA populations. Our analysis revealed significant genetic correlations between oil and protein content across all populations studied. The calculated values were as follows: -0.71 (A population), -0.75 (B population), -0.65 (C population), -0.50 (D population), -0.38 (E population), -0.26 (F population), and -0.87 (breeding population).

These findings are detailed on page 4, lines 158-162, and provide valuable insights into the genetic relationships between oil and protein content across different populations.

  1. Can you estimate the accuracy of the models?

Response: Thank you for your insightful comment. In our study, we defined prediction accuracy, or prediction ability, as the Pearson correlation (r-value) between the observed oil/protein content values or BLUEs (Best Linear Unbiased Estimates) and the Genomic Estimated Breeding Values (GEBVs) obtained from each genomic prediction model. While we attempted to assess prediction accuracy using this method, we acknowledge that determining the exact accuracy of the models remains a challenge.

We appreciate your suggestion and will explore additional methods for estimating model accuracy to enhance the robustness of our findings.

  1. The validation/testing set is a population resulting from the cross of two cultivars with oil and protein contents not very contrasting (the values are more or less similar?). Also, the oil and protein content of this population was measured in one year only, these quality traits are described to be highly influenced by the environment. This should be discussed.

Response: Thank you for your insightful comments. The parental soybean cultivars in our breeding population, JD12 and NF58, exhibit contrasting oil and protein contents. Specifically, JD12 is characterized by high protein content (~46%) and moderate oil content (~17%), while NF58 displays high oil content (~24%) and slightly lower protein content (~36%). The analysis of variance for oil and protein content between JD12 and NF58 yielded significant results (P < 0.001), underscoring the differences between these cultivars (see page 14, lines 530-533). We also acknowledge the influence of environmental factors on soybean oil and protein content. As the public data utilized in our study were sourced from a single environment, it is crucial to discuss the potential impact of environmental variation on the predictive ability of qualitative traits. We have incorporated this discussion into the manuscript's discussion section (page 13, lines 503-514). Thank you for bringing these important considerations to our attention.

Sincerely,

Chenhui Li

Round 2

Reviewer 2 Report

Comments and Suggestions for Authors

The authors answered back fairly. I do not have further questions. Some text editing is still needed.

Comments on the Quality of English Language

Some text editing is still needed.